# Co-Spray Drying of Paracetamol and Propyphenazone with Polymeric Binders for Enabling Compaction and Stability Improvement in a Combination Tablet

**DOI:** 10.3390/pharmaceutics13081259

**Published:** 2021-08-14

**Authors:** Ioannis Partheniadis, Ioannis Nikolakakis, Constantinos K. Zacharis, Kyriakos Kachrimanis, Nizar Al-Zoubi

**Affiliations:** 1Laboratory of Pharmaceutical Technology, Department of Pharmaceutical Technology, School of Pharmacy, Faculty of Health Sciences, Aristotle University of Thessaloniki, 54124 Thessaloniki, Greece; ioanpart@pharm.auth.gr (I.P.); kgk@pharm.auth.gr (K.K.); 2Laboratory of Pharmaceutical Analysis, Department of Pharmaceutical Technology, School of Pharmacy, Faculty of Health Sciences, Aristotle University of Thessaloniki, 54124 Thessaloniki, Greece; czacharis@pharm.auth.gr; 3Department of Pharmaceutics and Pharmaceutical Technology, Faculty of Pharmaceutical Sciences, The Hashemite University, 13133 Zarqa, Jordan; nzoubi@hu.edu.jo

**Keywords:** combination therapy, spray drying, experimental design, thermal analysis, compressibility, diametrical loading

## Abstract

Paracetamol (PCT) and propyphenazone (PRP) are analgesic drugs that are often combined in a single dosage form for enhanced pharmacological action. In this work, PCT and PRP were co-spray dried separately with hydroxypropyl methylcellulose (HPMC) and hydroxypropyl cellulose (HPC) using drug suspensions in polymer solutions as feed liquids. It was thought that because of polymer adherence to the surface of drug particles, the risk of PCT–PRP contact and interaction could be reduced. Such interaction may be caused by localized temperature gradients due to frictional forces during tableting, or during storage under harsh conditions. A worst-case scenario would be eutectic formation due to variations in powder mixture homogeneity since eutectic and therapeutic mass PCT/PRP ratios are close (65:35 and 60:40, respectively) and eutectic temperature is low (~56 °C). Uniform particle size, round shape, compaction improvement and faster release of the analgesics were important additional benefits of co-spray drying. Experimental design was first applied for each drug to optimize the polymer concentration on the yield of spray drying and melting point separation (Δmp) of heated binary mixtures of co-spray dried PCT/neat PRP, and vice versa, with the two drugs always included at their therapeutic 60:40 ratio. Optimal combinations with largest Δmp and production yield were: co-spray dried PCT (15% HPC) with neat PRP and co-spray dried PRP (10% HPMC) with neat PCT. Compression studies of these combinations showed tableting improvement due to the polymers, as reflected in greater work of compaction and solid fraction, greater fracture toughness and tablet strength, easier tablet detachment from the punch surface and ejectability. Faster release of both drugs was obtained from the tablet of co-spray dried PCT (15% HPC) with neat PRP. A one-month stability test (75% RH/40 °C) showed moisture-induced alteration tablet strength.

## 1. Introduction

Fixed-dose drug combinations in a single dosage form can provide more therapeutic or pharmacological value than each drug separately, as well as ensuring production cost savings. Additionally, patient compliance is improved if two drugs are taken as a single combinatory dosage form instead of two separate medications, since this will reduce the administration frequency. Among the various drugs, analgesics and antipyretics are very popular as prescription or over-the-counter medicines alone or in combination products. Paracetamol (PCT) is probably the antipyretic with the largest circulation and propyphenazone (PRP) is another analgesic that is commercially available as a combination product with PCT. A triple-combination tablet containing PCT, PRP and caffeine is also commercially available under the trade name Saridon^®^. These combinatory treatments aim to enhance pharmacological action by drug synergy. On the technological side, PCT exhibits serious tableting problems due to the inherent poor compressibility and high elastic recovery which has been the subject of many studies [1,2,3]. On the other hand, similar problems of poor compressibility, flowability and tabletability of PRP have received little attention. In one study, improvement of flow and compression of PRP was achieved by spherical crystallization [4].

Spray drying with suitable adjuvants is an effective strategy that has been applied to convert poorly compressible drugs into powdered products capable of direct compression [5]. This is especially desirable for high-dose drugs where the large amounts of excipients required for tableting may lead to oversized tablets. Moreover, encapsulation of incompatible ingredients by spray drying can be used to combine them into a single fixed-dose tablet form [6]. By making a correct choice of encapsulating materials, favorable alterations of particle surface morphology and micromeritic properties can be made resulting in good flowability and compressibility [6,7,8,9]. Attempts to improve the properties of PCT by co-spray drying with various excipients have been already reported in the literature [9,10]. In particular, McDonagh et al. [9] investigated the role of the soluble fraction of α-lactose monohydrate on the tableting of their crystallo-co-spray dried agglomerates. They reported that by increasing the soluble fraction of lactose, the interaction with PCT became more effective, resulting in greater compressibility and tabletability.

Regarding the stability of drugs in combination dosage forms and the risk of interaction, it has been reported that due to the applied compressive stresses during compaction, intimate contact of mutually soluble eutectic-forming compounds may result in melting of them as a single phase at a temperature lower than the melting point (mp) of each one of them [11]. Moreover, from infrared measurements of tablet surface temperature and application of finite element analysis, it is known that frictional forces between particles and between the powder and machine tools may cause temperature rise at the edges of tablets above 44 °C which for non-compressible materials at high compaction speeds can be even higher due to increased friction [12,13]. In conclusion, the combined action of released heat and compressive forces may result in drug–drug interaction and eutectic melting with implications on the solubility, dissolution, mechanical properties, solid state and stability. This is a serious cause of concern in the tableting of PCT and PRP, since their eutectic composition is 65:35 PCT/PRP which is close to therapeutic 60:40, at relatively low temperature of 56 °C [11,14]. Moreover, physicochemical instability of the combination PCT/PRP tablet when stored for one month at 40 °C and 75% relative humidity has also been reported [14].

Therefore, in the case of the PCT with PRP combination tablet there is high risk of interaction of the mutually soluble drugs and even eutectic formation due to unforeseen local variations in powder mixture homogeneity, high compressive stresses and temperature rise. Interaction may also occur during storage at high temperature/humidity conditions. One simple way to reduce the risk of interaction is to prevent drug–drug intimate contact in the dosage form. This could be achieved by spray drying a drug suspension in a solution of a functional polymeric excipient with adhesive and binder properties, so that during the short co-spray drying time, the polymer could form a protective layer on the particle’s surface. Additionally, due to the binder ability, it could promote agglomeration of primary drug particles into spherical agglomerates of uniform size, giving them better packing ability and compactibility.

Design of Experiments (DoE) is extensively used for implementation of Quality by Design platform (QbD) aiming to better understand the process and assure product quality. Statistical models are established by correlating the experimental factors with the measured variables. Mathematical predictive equations are produced from the models that enable the use of desirability concept. Desirability is quantified with the aid of numerical optimization tools offered by software programs in a zero-to-one scale as a measure of optimization success of a certain response. Subsequently, the individual desirabilities of the measured responses can be condensed into an overall desirability reflecting the best combination of responses and experimental factors as well [15].

Therefore, in the present study, PCT and PRP were co-spray dried with hydroxypropyl methylcellulose (HPMC) and hydroxypropylcellulose (HPC) which have strong adhesive properties that have established them as film-forming agents in solid dosage forms. The purpose of the work was two-fold: (i) to create a polymer barrier around the drug particles to minimize PCT–PRP contact and interaction and (ii) to produce spherical particles with uniform particle size and improved compressibility, thus enabling formation of a direct compression combination tablet with PCT over PRP mass ratio 60:40 [16,17]. The work was divided into two parts. Firstly, suspensions of each drug were co-spray dried with aqueous polymer solutions of different concentrations (5, 10 and 15% w/w) to give six co-spray dried (CSD) batches for each drug. Optimization was applied for each drug to select the polymer type and content that maximized production yield and melting point difference (Δmp) when PCT-CSD with neat PRP and PRP-CSD with neat PCT binary mixtures of drugs were heated together. Next, mixtures of the two optimal combinations were compressed on an instrumented tablet press and, “in-die” parameters characteristic of compressional behavior were derived. Diametrical loading fracture tests of the tablets were also performed to estimate fracture toughness and tensile strength. In addition to reflecting the ability of the tablets to withstand rigors during handling and processing, the last two parameters reflect the feasibility of splitting the tablet into smaller doses if needed. Lastly, the optimal tablet combinations were subjected to accelerated stability test according to the ICH Guidelines. Raman spectra of the tablets were supplementarily taken before and after the stability test to enlighten possible interactions associated with changes in tablet properties.

## 2. Materials and Methods

### 2.1. Materials

Propyphenazone powder (Eur. Ph. grade from Medex, Rugby, UK) was donated kindly by MID PHARMA (Amman, Jordan) and paracetamol (PCT) monoclinic fine powder was donated by Boehringer Ingelheim (Ingelheim, Germany). HPC (H grade) was obtained from Nisso (Nippon SODA Co. LTD, Tokyo, Japan) and HPMC (Methocel K4M) from Colorcon (Dartford Kent, UK). Methanol (HPLC grade) and concentrated H_3_PO_4_ (85 % *w*/*w*) were from Sigma-Aldrich (St Louis, MO, USA). High-purity water (18.2 MΩ cm resistivity) that was used for the HPLC analysis was produced by a B30 water purification system (Adrona SIA, Riga, Latvia). PCT had a mean particle size of 27 μm (span, 2 μm) and was used as received. PRP was fractionated using an Alpine Zig-Zag classifier (Augsburg, Germany) and the size fraction separated at a speed of 6000 min^−1^ (corresponds to 25 μm) with mean particle of 16 μm (span, 1 μm) was used.

### 2.2. Methods

#### 2.2.1. Co-Spray Drying

Each drug was co-spray dried separately. Firstly, solutions of the two polymers in distilled water were prepared at three concentrations: 0.5, 1.0 and 1.5% *w*/*v*. Then, 300 mL of each solution was added slowly to 30 g of each drug in a 600 mL beaker to form drug suspensions at three polymer/drug mass percentages: 5%, 10% and 15%. The suspensions were stirred magnetically until complete de-agglomeration (for about 20 min) and were used as the feed liquids. For spray drying, a Büchi B-191 mini spray dryer (Büchi Labortechnik AG, Flawil, Switzerland), fitted with 0.7 mm nozzle was used at inlet temperature 120 °C, feed rate 6 mL/min, airflow 800 L/h and aspiration 35 m^3^/h. After all feed liquid was sprayed, heating was stopped while aspiration continued until the inlet temperature dropped below 70 °C. The sample was collected, weighed and kept in screw-capped bottles until required for testing.

#### 2.2.2. Loading Efficiency

Samples of about 10 mg of each co-spray dried (CSD) batch were dissolved in 100 mL distilled water with the aid of an orbital shaker (SO1, Stuart Scientific, Stone, UK). The loading efficiency was determined by HPLC—UV analysis after appropriate dilution. The procedure was carried out in triplicate and the mean and standard deviation were calculated.

#### 2.2.3. HPLC—UV Analysis

Liquid chromatography was applied to analyze samples taken during dissolution testing and determination of loading efficiency. A Shimadzu HPLC system consisting of two LC-20AD high pressure gradient pumps, an SIL-20AC HT thermostated autosampler, a CTO-10ASVP column compartment and SPD-20A PDA detector (Kyoto, Japan) was employed. The instrument control and the data handling were carried out via LC Solutions software (Kyoto, Japan) (version 1.25 SP4). An analytical column Supelcosil LC-8 (150 × 4.6 mm id, 5 μm) (Supelco) was employed in all cases. The mobile phases A and B consisted of 0.1% *v*/*v* aqueous solution of phosphoric acid and methanol, respectively. The gradient elution steps comprised: an initial content of 20% A for 2 min, linear increase up to 100% in 8 min that was kept constant for 1 min and then changed to 20% B within 1 min followed by 10 min column equilibrium to obtain reproducible retention time. Flow rate was 1 mL/min, injection volume 20 μL and the column temperature 40 °C. Both drugs were monitored spectrophotometrically at 243 nm. Samples were kept at 15 °C in the autosampler tray. The analytical method was validated according to ICH guidelines. The specificity of the method was assessed by performing forced degradation studies (under hydrolytic, acidic, basic, oxidative, thermal and photolytic conditions [18]. The results showed that the developed HPLC method is specific and, in all cases the peak purity index of both drugs was higher than 0.99999 (Appendix A). The retention times of PCT and PRP peaks remained unaffected during the stressed studies and the degradation products were well-resolved from the analytes’ peaks. Linearity was tested between 0.1 and 20 μg mL^−1^ using seven calibration levels of each drug. In all cases, the correlation coefficients were >0.9999. Residual plots indicated random distribution around “zero” line. Repeatability was <0.5% of the relative standard deviation of six replicates of standard solution of each analyte (1 mg/L). Lower limits of quantification (LLOQ) were 100 ng/mL for both drugs.

#### 2.2.4. ATR-FTIR Spectroscopy

FTIR spectra of neat and co-spray dried (CSD) drugs were recorded using a Shimadzu FT-IR-Prestige-21 spectrometer (Shimadzu Corporation, Tokyo, Japan) supported by software (Shimadzu IR solution 1.3, Shimadzu Europa GmbH, Duisburg, Germany) to see if there are any interactions between drugs and polymers. The instrument was attached to a horizontal Golden Gate MKII single-reflection ATR system (Specac, Kent, UK) equipped with a Diamond/ZnSe crystal (45° angle to the beam, 1.66 μm at 1000 cm^−1^ depth of penetration, 2.4 refractive index, and 525 cm^−1^ long wavelength cut-off). Samples were placed on the diamond disk using a sapphire anvil to restrain the powder. The spectra were average of 64 scans collected at 4 cm^−1^ resolution.

#### 2.2.5. Differential Scanning Calorimetry (DSC) Analysis

Thermal analysis of CSD drugs alone and in binary mixtures with neat drugs was performed to identify thermal changes due to co-spray drying and due to their copresence in the mixture at the therapeutic PCT/PRP ratio 6:4. A Shimadzu DSC-50 differential scanning calorimeter (Shimadzu Corporation, Kyoto, Japan) connected to Shimadzu TA-60WS Thermal Analyzer (Version 2.21, Shimadzu Corporation, Columbia, MD, USA) was used for data acquisition and processing. Samples of 4–5 mg were weighed, sealed into aluminum pans, and heated from 25 °C to 200 °C at a rate of 5 °C/min under nitrogen atmosphere.

#### 2.2.6. Scanning Electron Microscopy (SEM)

SEM photomicrographs of plain and CSD particles were taken with a scanning electron microscope (JEOL JSM-6390LV, Tokyo, Japan) and their outline and surface morphology examined. The samples were placed on double-sided electro-conductive adhesive tape, which was fixed on aluminum stubs and then sputter-coated with carbon (20 nm thickness). Microphotographs were taken at 20 kV.

#### 2.2.7. Work of Compression, Elastic Recovery, Detachment, and Ejectability of Tablets

Twenty-five milligrams of binary CSD/neat drug mixtures at PCT to PRP mass ratio 60:40) of optimal composition (polymer type and content, Section 2.2.11) were compressed using an instrumented press (Gamlen D-Series Press, Nottingham, UK) fitted with Ø4.0 mm flat-edged punches, at 156 MPa pressure and 10 mm/min speed. Force-displacement (f/d) profiles were recorded during compression, detachment and ejection. From these, the work of compression (*W**_c_*, Equation (1)) and elastic recovery (*ER*%, Equation (2)) that characterize compression behavior were obtained. *W**_c_* was obtained from Equation (1) as the area enclosed by the compression/decompression curve. *ER%* was obtained from Equation (2) as the %increase in tablet thickness after removal of compressive force (*p* = 0) relative to the thickness at maximum punch penetration.
(1)$$W_{c}=\int_{0}^{h}dx-\int_{h}^{h^{\prime }}dx$$
(2)$$ER\;\%=\frac{h^{\prime }-h}{h}\;\times \;100$$
where *h* is the compact thickness at *p* = 0 and *h*’ at maximum punch penetration. From the ejection profiles, ejectability was estimated as maximum ejection force. Additionally, the force required to detach the tablet from the base plate (which had the role of lower punch) due to sticking was measured by turning the die at 90 degrees to compression position and applying force to the base plate at 10 mm/min.

#### 2.2.8. Solid Fraction

For the determination of solid fraction (*p*_F_) three tablets of each optimal combination were tested. Their weight and thickness were measured using analytical balance (0.001 g accuracy, UW 220 Shimadzu, Japan) and micrometer (0.01 mm accuracy, Moor and Wright, Sheffield, UK). Solid fraction (*p*_F_) was then calculated from Equation (3):*p*_F_ = *k* ∗ *w*/*h*_t_(3)
where *w* is the tablet weight (mg), *h*_t_ its thickness (mm) and *k* the numerical coefficient incorporating the tablet diameter (Ø 4 mm) and true density of the tablets calculated from the true densities of the components using literature values (for HPMC 1.326 g/cc, HPC 1.222 g/cc, PCT 1.214 g/cc, PRP 1.1407 g/cc) [4,9,19]. For PCT/PRP, *k* was 0.0672, for PCT-CSD/PRP 0.0671 and for PRP-CSD/PCT it was 0.0668.

#### 2.2.9. Fracture Toughness, Young’s Modulus and Tensile Strength

Fracture profiles under diametrical loading were obtained using the instrumented press described above for compression but operated in fracture mode using a 10 kg load cell. Loading was applied by the upper moving head at 1 mm/min. Fracture toughness or work of failure (*W**_f_*), representing the energy needed to break the tablet, was obtained from the area enclosed by the fracture curve, the line drawn perpendicular to the break point (*h**_f_*) and the distance on X-axis from the intersection of the perpendicular line to the origin of X-axis according to Equation (4):(4)$$W_{f}=\int_{0}^{h_{f}}dx$$

Elastic deformation influences the stresses in the tablet that may cause breaking of bonds and defects and thus, reduce fracture toughness and strength [20]. It was estimated as the apparent Young’s modulus obtained from the slope (d_stress_/d_strain_) of the initial linear part of the load–displacement fracture curve. It is an index of the rigidity or the resistance of the tablet to deform axially under compressive load.

Tensile strength (*TS*) is an important attribute of tablet quality resulting from the synergy of plastic deformation and inter-particle bonding. It takes into account the tablet geometry and its measurement and calculation are described in the United States Pharmacopoeia (USP <1217>) [21]. A minimum value of 1 MPa practically defines a useable tablet, but a target of ≥2.0 MPa is a criterion for acceptable friability [22]. Mechanical strength was evaluated as fracture tensile strength under diametrical loading *TS* (Mpa) and was calculated from Equation (5) [23]:*TS* = 2*F*/*πΦh*_t_(5)
where *F* is the breaking load (N), *Φ* the tablet diameter (mm) and *h*_t_ its thickness (mm).

#### 2.2.10. Stability Study

To evaluate the stability of the combination tablets of the optimal binary mixtures of PCT-polymer/PRP and PRP-polymer/PCT under extreme environmental conditions, a one-month accelerated test was conducted following the ICH Guidelines. According to these guidelines, tablets were exposed to 75% relative humidity (desiccator with saturated NaCl solution) and 40 °C for one month. Since the test was carried out on unpacked tablets exposed to conditions that mimic the worst-case scenario of storage, we consider that the one-month period is adequate. Raman spectra of the entire tablets were taken before and after the stability test and were related to changes in the fracture parameters. The % drug content of the tablets at the end of the stability test was determined by the developed HPLC-UV method (see Section 2.2.3). A physical mixture of neat drugs was also tested for comparison. Tablets contained PCT over PRP at mass ratio 60:40.

#### 2.2.11. Raman Spectroscopy

Raman spectroscopy was used to examine changes in the tablets of the optimal binary mixtures subjected to stability test. It was used instead of FTIR for several reasons: because it has been used to study the structure and interactions of the two studied drugs [24], because it allows spectra acquisition of the entire intact tablet without the need to break them, and because the presence of moisture in the tablets does not obscure interactions between the ingredients. Spectra were recorded using a bench top Raman spectrophotometer (Agility, dual band 785/1064 nm model, BaySpec, San Jose, CA, USA) supported by suitable software (Agile 20/20). Tablets were griped in the special holder and the laser beam was focused on the tablet surface with the aid of a fine adjustment knob. Spectra were recorded in the wavelength range 100–2700 Raman shifts (cm^−1^) at resolution 12 cm^−1^, exposure time 10 s and power of incident laser beam 250 mW, using excitation line 785 nm. Recorded spectra were average of 10 runs.

#### 2.2.12. In Vitro Dissolution Study

In vitro release studies were performed using USP Apparatus II (paddle) system (Pharma Test PTW 2, Hainburg, Germany) at 100 rpm. A tablet of the optimized combinations or physical mixture was added to 900 mL of PBS solution (pH 6.8) at 37 ± 0.5 °C. Aliquots were taken at timely intervals from 5 to 150 min. Due to the overlap of the peaks of the two drugs, UV spectroscopy could not be used for the analysis of the two drugs in the combination tablet. Instead, chromatography was applied for the simultaneous analysis of the dissolution samples (see Section 2.2.3). Representative HPLC-UV chromatographs of samples taken at various time intervals during the test are shown in Figure 1.

The drug release data were described using the Weibull equation (Equation (6)) [25,26]. It distinguishes drug release mechanisms, which in the present work is very likely due to the different release characteristics of HPMC and HPC tablets [27].
ln[−ln(1 − *W*/*W*_0_)] = −ln*a* + *b*ln(*t* − *t*_0_)(6)
*W* is the drug released at time *t*, *W*_0_ is the drug released at the end of the test, *t*_0_ is the lag time before release as determined by trial and error for best line fitting, *b* is constant characteristic of the shape of the release curve and release mechanism [28], and *a* is a time-scale parameter defined as *a* = (*t*_d_)*b* where *t*_d_ is the time required for 63.2% release.

#### 2.2.13. Experimental Design and Optimization

A mixed-level factorial design with polymer type at two levels (HPMC or HPC) and polymer content at three (5%, 10%, 15% relative to drug mass) was applied separately for each drug. The six experimental points were replicated, and three more replications of one experimental point were added for lack of fit, giving a total of 15 experimental points for each drug design. ANOVA was applied to evaluate the mathematical equations derived using two factor-interaction models that quantified the dependence of spray drying yield and melting endotherm’s separation (Δmp) in CSD/neat drug mixtures on the studied factors. The derived equations were used for numerical optimization for maximum production yield and Δm. The optimal CSD/neat combinations were further progressed into compression/fracture/release/stability studies, as shown in the flow chart in Figure 2. The results were expressed as partial desirabilities of yield production and Δmp, and the overall desirability *D* was calculated. Compositions with highest *D* were used for compression/fracture/release and stability. In all experiments, the mass ratio in the mixtures was paracetamol 60, propyphenazone 40.

The Equation (7) for production yield and Δmp maximization was:*d*i = (*Y*_i_ − *Y*_min_)/(*Y*_max_ − *Y*_min_)(7)
where *d*i is the desirability function of a response ranging from 0 to 1, *Y*_min_ is the lowest measured value, *Y*_max_ is the highest measured and *Y*_i_ is any value. When the response value is outside the desired range *d*i = 0 and when it is within is *d*i = 1. The overall desirability *D* for each drug was the geometric mean of the individual desirabilities (Equation (8)):*D* = (*d*1 × *d*2)^0.5^(8)
where *d*1 is the partial desirability for yield and *d*2 for Δmp. Design Expert 8.0 (Stat-Ease, Minneapolis, MN, USA) was used to generate the experimental design and perform statistical analysis and numerical optimization.

## 3. Results and Discussion

In all binary mixtures of drug combinations and tablets made from them, the PCT to PRP mass ratio was 60:40 regardless of which drug appears before or after the slash (/) in the abbreviated mixture notation.

### 3.1. Co-Spray Drying, Loading Efficiency and Melting Point

Experimental batches and codes, together with production yield and loading, are presented in Table 1. For PCT co-spray dried with HPMC, the yield ranged from 10.7 to 18.6% and with HPC from 9.6 to 19.2, whereas for PRP co-spray dried with HPMC, it ranged from 11.0 to 19.8% and with HPC from 7.8 to 18.9%. There was no significant difference between the effects of the two polymers. Increasing polymer content proportionally decreased the yield for both drugs. This is ascribed to the adhesiveness and the increased presence of the polymers at the droplet surface as their content increases, resulting in greater sticking to the drying chamber wall. Furthermore, from Table 1 it can be seen that the loading efficiency decreased with polymer content indicating that during spray drying polymer replaced some drugs in the agglomerates which were drifted away from the aspiration air current due to their finer size. Overall, the yield values were lower compared to those previously found [29], which is attributed to the higher polymer content (5 to 15% relative to drug) employed in this work, compared to 2.5% to 5% in the previous.

### 3.2. ATR-FTIR Analysis

In Figure 3, ATR–FTIR spectra of PCT (Figure 3a) and PRP (Figure 3b) neat and CSD drugs with 15% HPMC and 15% HPC are presented. Spectra of samples with highest (15%) added polymer were recorded so as to ensure that any interactions would be revealed. From Figure 3a, it can be seen that for neat PCT, characteristic peaks appear at 1650 cm^−1^ (C=O stretching band), at 1506 and 1562 cm^−1^ (C-N amide stretching) and at 1614 and 1440 cm^−1^ (C-C aromatic stretching) [24,30]. These also appear in the spectra of PCT-hpc15 and PRP-hpmc15 in Figure 3a without sh0ift, indicating lack of drug–polymer interaction. From Figure 3b, it can be seen that in the spectra of PRP there are peaks at 1652 cm^−1^ (C=O amide stretching) and at 1614 and 1453 cm^−1^ (C-C aromatic stretching) which also appear in the spectra of PRP-hpc15 and PRP-hpmc15 powders at the same position. However, peaks in the regions 1000–1050 cm^−1^ and 1150–1250 cm^−1^ of the PRP-hpc15 spectrum appear suppressed (arrow) implying some interaction.

### 3.3. Thermal Analysis

In Figure 4a, DSC thermograms for co-spray dried PCT with HPMC or HPC and in Figure 4b thermograms of PRP with HPMC or HPC are presented. In Figure 5, thermograms of binary mixtures of PCT-CSD with neat PRP (Figure 5a) and PRP-CSD with PCT (Figure 5b) are presented. Melting points (mps’) of CSD drugs alone and in binary CSD/neat mixtures are given in Table 2.

The thermograms in Figure 4 show clear melting peaks of CSD drugs alone. There is some shifting from the measured mp of neat PCT (170.3 °C, form I) (Figure 4a, dotted line) and neat PRP (103.2 °C, form II) (Figure 4b, dotted line). From Table 2, it appears that the mps’ for PCT co-spray dried with HPMC were between 166.4 and 170.5 °C and with HPC between 164.9 and 171.3 °C which are reasonably close to the mp of pure drug (170.3 °C). The small differences can be mainly attributed to variation in the degree of aggregation and in the distribution of polymers on the particles that may influence thermal conductivity and DSC measurement [11]. Furthermore, from Table 2, it can be seen that the mps’ of PRP-hpmc are between 101.3 and 104.1 °C and those of PRP-hpc between 103.1 and 103.2 °C, which are quite close to that of the pure drug (103.2 °C). The lower variability and closer mps’ of co-spray dried PRP powders to pure drug is explained by the lower mp of PRP compared to polymer’s glass transition temperature (Tg) (for HPMC Tg it is 170–180 °C and for HPC 130 °C [19]). Therefore, during DSC heating, the polymers remain solid and fixed on the PRP particle surface. On the other hand, since the mp of PCT (170.3 °C) is higher than the Tg of the polymers, fluidization and redistribution of polymers during melting of PCT is possible, causing variation in the recorded mps’ of PCT-CSD powders.

In Figure 5, DSC thermograms of binary physical mixture (PCT/PRP) and of PCT-CSD/PRP (Figure 5a) and PRP-CSD/PCT (Figure 5b) mixtures are presented. Two separate melting peaks are seen located at much lower temperatures compared to the mps’ of CSD drugs heated alone (Figure 4), indicating PCT–PRP interaction and structure weakening. It is particularly noticeable that the well-formed melting peak of PCT-CSD in Figure 4a has been reduced to a shallow endotherm in the physical mixture in Figure 5a (top curve). Since PCT has H-bond hydroxyl donor and PRP carbonyl accepting group, H-bonding should be responsible for the interaction. A second heating cycle of the physical mixture performed after one day showed that drugs remained amorphous.

Comparing the position of melting endotherms in the thermogram of the physical mixture (Figure 5a,b, top curve) with those of CSD/neat mixtures (below), it appears that the endotherms in the latter are shifted towards higher temperatures. This signifies inhibition of intimate contact between the two drugs and less interaction due to the polymer barrier on the particle surface. Furthermore, from the mps’ of the drugs in the binary mixtures (Table 2), it appears that the presence of polymer resulted in greater mp increase for PRP than PCT. For the PCT-CSD/PRP mixtures, the increase in the mp of PCT relative to its mp in the physical mixture (132 °C, Figure 5a top) was between 7.0% and 17.0%, whereas for the PRP-polymer/PCT mixture, the increase in the mp of PRP relative to neat drug in the physical mixtures (61.0 °C, Figure 5a) was between 15.6% and 46.4%. This means that PRP was better protected by the polymers. An explanation can be offered based on the Tg of polymers relative to the melting points of neat drugs. The Tg values of HPMC (around 170–180 °C) and HPC (around 130 °C) are higher than the mp of PRP (103.2 °C), but below or near the mp of PCT (170.3 °C). Thus, during DSC heating, the polymers remain as solid protective layers fixed on the surface of PRP particles, but become mobile, fluid and less adherent on the PCT particles, offering less protection from contact with PRP.

It is interesting to notice from Table 2 that the effect of polymer on the mps’ of the drugs in the binary mixture is not exerted only on the CSD drug but also on the drug present as neat. For example, for PCT, the increase in mp in its CSD form relative to the mp in the physical mixture (133.0 °C) was between 5.9% and 27.4%, while for the PCT present as neat was between 16.1% and 38.5% (Table 2, upper half, columns 4, 6). Similarly, for PRP in its CSD form, the mp increase relative to the mp in the physical mixture (61 °C) was between 15.6% and 46.4%, while for the PRP present as neat it was between 11.0% and 23.6% (Table 2, lower half, column 4, 6). This finding signifies that the protective polymer layer on the particles of one drug simultaneously provides protection for the other by prohibiting contact. This result is of practical significance since it leaves an open choice to the formulator as to which of the two drugs to process as CSD, considering improvements in other properties, such as mechanical and release.

Since one of the main objectives of this work was to find the optimal co-spray drying conditions (polymer and content) for each drug for the greatest mp separation (Δmp) in the CSD/neat drug mixtures, the dependence of Δmp on the polymer and its content was examined. In Figure 6, plots of Δmp against polymer content are depicted for the four drug/polymer experimental groups (Table 1). Except for mixtures PCT-hpmc15/PRP, PCT-hpc5/PRP and PRP-hpmc15/PCT, Δmp is greater than that of the physical mixture (Δmp = 71.5 °C, dotted line). There is no certain trend of change except for the PCT-hpc/PRP which follows a straight-line increase. This implies firm attachment of HPC on the PCT particles and increasing surface coverage as more polymer is added. Their good affinity can be attributed to the H-bonding of PCT with HPC by virtue of their ability to act both as donors and acceptors of hydrogen.

### 3.4. Analysis of DoE and Numerical Optimization

ANOVA results with fitting indices derived from the analysis of the experimental design are presented in Table 3. The factors were: the polymer type (X1) and its content relative to drug (X2) and the measured variables were the production yield and the Δmp of the CSD/neat drug mixtures. From Table 3, it can be seen that in all cases, the two interaction-factor regression model (containing the term X1 × X2) fitted the data very well (there was no lack of fit). Both examined factors significantly affected yield and Δmp and there is interaction of their effects (*p* = 0.026 and 0.027 for yield and <0.001 for Δmp). For Δmp, the different dependence of Δmp on polymer type is clearly seen in Figure 6.

Νumerical optimization was performed to maximize production yield and mp difference (Δmp) of the drugs in their heated CSD/neat binary mixtures. The results are presented in Table 4 as desirability values for production yield (4A), Δmp of PTC-CSD/PRP (4B) and Δmp of PRP-CSD/PCT (3C). Considering production yield, high desirabilities between 0.919 and 0.983 (scale 0 to 1) were obtained for both drugs and polymer contents, except for PRP co-spray dried with 15% HPMC (PRP-hpmc15). Considering Δmp, desirability values above 0.9 were obtained for PTC-hpc10, PCT-hpc15 and for PRP-hpmc10. From the partial desirabilities of production yield and Δmp, the overall desirability (*D*) was calculated for each drug. The solutions of highest *D* pointed to processing PCT with 15% HPC (batch PCT-hpc15, Table 1) and PRP with 10% HPMC (batch PRP-hpmc10, Table 1). These batches were subsequently used to make tablets from the corresponding CSD/neat binary mixtures which were studied for compression, ejection, fracture and stability performance, as shown in the flow chart of work progress in Figure 2.

### 3.5. SEM Microphotographs of Optimal Co-Spray Dried Drug/Polymer Compositions

In Figure 7, SEM microphotographs of neat drugs and the optimal CSD batches PCT-hpc15/PRP and PRP-hpmc10/PCT of the two drugs are presented. It can be seen that co-spray drying transformed the particles from irregularly shaped to round agglomerates of more or less uniform particle size of less than 30 microns. This signifies the ability of both HPMC and HPC to adhere onto drug particles during spray drying, and form layers and interparticle bridges leading to agglomeration. The considerably lower particle size of co-spray dried PRP compared to neat PRP particles is because the small size fraction of the fractionated supplied PRP was used for co-spray drying.

### 3.6. Mechanical Properties of Tablets

Both polymers that were employed in this work are known tablet binders that can be deposited on the particles during co-spray drying. Therefore, they are expected to improve any surface-related compaction parameters of the powders of the two optimal CSD/neat drug combinations and of the mechanical properties of the corresponding tablets, in addition to prohibiting direct PCT–PRP contact.

#### 3.6.1. “In-Die” Measured Parameters of Compressed Powder

‘In-die’ measured parameters as indices of compression performance can contribute to an in-line monitoring strategy. Figure 8 presents f/d profiles recorded during compression/decompression, detachment and ejection of optimal binary PCT-hpc15/PRP and PRP-hpmc10/PCT mixtures. Profiles for the physical mixture (PCT/PRP) of drugs is also included for comparison. Values of derived “in-die” parameters of work of compaction (*W**_c_*), detachment force (*F*_det_) and ejection force (*F*_ej_) are given in Table 5 together with elastic recovery.

Work of compaction represents the ability of a materials to absorb compaction work through interparticle surface interaction and rearrangement. From the values in Table 5, it appears that the *W**_c_* of physical mixtures is smaller (569.0 kJ) than the *W**_c_* of PCT-hpc15/PRP (664.2 kJ) and PRP-hpmc10/PCT (612.6 kJ) tablets. Since neat and CSD powders were more or less in the same size range (16 to 27 μm median diameters; see Materials, Section 2.1 and Figure 7), the higher *W**_c_* of the CSD/neat drug mixtures should be due to the presence of polymers on the particles, increasing deformability, contact area and extent of bonding [31].

Elastic recovery (*ER*) is associated with loss of interparticle bonds during decompression and loss of tablet strength. From Table 5, it can be seen that *ER* remained high after co-spray drying, between 25.2 to 29.5%. This was expected since *ER* is a bulk material property, whereas polymers only modified the particle’s surface.

Detachment of the tablet from the punch surface is not usually measured during tableting. However, for materials that exhibit capping and lamination, as in the case for the present drugs, during detachment shearing forces develop that may be greater than those the tablet can resist, resulting in capping and lamination [32]. From Figure 8b, it can be seen that clear detachment profiles were recorded from which the peaks recorded at the beginning of force application were taken as the detachment force (*F*_det_). From Table 5, it appears that tablets of physical mixture exhibited higher *F*_det_ (59.8 N) than PCT-hpc15/PRP (48.9 N) and PRP-hpmc10/PCT (50.2 N). Since polymers themselves increase adhesion, the reduced *F*_det_ of the CSD/neat tablets is due to their greater coherence.

Ejection force (*F*_ej_) provides an estimate of frictional and adhesive forces between the exiting tablet and the die wall, which may cause scratches on the sides and chipping of the edges of the tablets. It is therefore closely related to manufacturability. Although lubricants are added to ameliorate such problems, they may impact on the dissolution and tablet strength. For this reason, ejectability of the unlubricated tablet is important as an indication of further lubrication requirements. From Figure 8c, it can be seen that clear ejection profiles were recorded. Peak values were taken as *F*_ej_ and are shown in Table 5. It can be seen that tablets of the physical mixture exhibited higher *F*_ej_ (162.3 N) than PCT-hpc15/PRP (117.4 N) and PRP-hpmc10/PCT (124.1 N). Since polymers increase adhesion, the reduced *F*_ej_ of the CSD/neat tablets should be due to their greater coherence.

#### 3.6.2. Solid Fraction

In Table 6, results of thickness, weight and calculated solid fraction *p*_F_ of the tablets before and after the stability test are presented. The tablets of physical mixture showed considerably lower *p*_F_ (0.893) compared to PCT-hpc15/PRP (0.955) and PRP-hpmc10/PCT (0.940) signifying an active role of polymers on densification. Firstly, the round shape of the agglomerated CSD particles promotes easier packing in the initial compression stage which assists closer positioning and more efficient operation of compressive forces during compaction as it was previously shown from correlations between powder packing indices and tablet strength [33]. Secondly, the greater deformability of the polymer-modified particle surface assists closer rearrangement and greater consolidation during compaction.

The exposure of tablets to the conditions of stability test (one month at 75% RH, 40 °C) resulted in *p*_F_ changes that differed for the three drug combinations. For the physical mixture (PCT/PRP) and PCT-hpc15/PRP tablets there was a significant *p*_F_ decrease (0.893 to 0.769 and 0.955 to 0.843, respectively), whereas for PRP-hpmc10 there was a small increase (0.940 to 0.946). These changes are better understood and interpreted if they are examined on the basis of mass and thickness changes which control *p*_F_. For the PCT/PRP tablets, there was a considerable thickness increase of 17.7% and small weight increase 1.1% resulting in 13.9% *p*_F_ decrease, and for PCT-hpc15/PRP tablets there was greater thickness increase of 18.6% and small weight increase 1.1% resulting in a 11.7% *p*_F_ decrease. On the other hand, for the PRP-hpmc10/PCT tablets, there was no thickness change and only a small 0.12% weight increase, resulting in a small 0.6% *p*_F_ increase.

Since the studied drugs exhibit insignificant moisture sorption [34], the *p*_F_ decrease in the physical mixture tablets is due to the increased elastic recovery due to disruption of weak bonds by the adsorbed moisture and axial expansion. For the CSD particles, the dimensional and mass changes should be due to uptake of moisture by the polymers. At 75% RH, both polymers contained about 8% moisture [19]. However, the impact of moisture was different between the PCT-hpc15/PRP and PRP-hpmc10/PCT tablets. In the former, the considerable thickness increase (1.03 to 1.23 mm) and *p*_F_ decrease (0.955 to 0.843) should be due to moisture-induced swelling of the polymer. This has been previously shown with HPC matrix tablets equilibrated at RH 70.4% and 93.9% levels [35]. Therefore, under the 75% RH of the stability test, the thickness increase and *p*_F_ decrease in the PCT-hpc15/PRP tablets are primarily due to swelling of the polymer. Since this is located around the drug particles, swelling will increase the overall interparticle distance and tablet thickness. The very small thickness increase and *p*_F_ decrease in PRP-hpmc10/PCT tablets implies that moisture is accommodated in the tablet without polymer swelling.

#### 3.6.3. Diametrical Fracture Test

In Figure 9, fracture profiles of breaking load against displacement of the compression head of the tester during diametrical loading are presented for tablets of physical mixtures (PCT/PRP) and CSD/neat drug mixtures. Solid lines represent green tablets (Day 0) and dotted lines aged tablets at the end of stability test (Day 30). Results of the analysis of the fracture curves in terms of toughness (or work of failure, *W**_f_* = area under the fracture curve), Young’s modulus (*E*) or tablet rigidity (=slope of the linear part of fracture load–displacement curve) and fracture tensile strength (Equation (5)) are given in Table 7. From Figure 9, it appears that all green tablets (solid lines, Day 0) gave clear break points. CSD/neat powder mixtures produced much stronger tablets than PCT/PRP accompanied with higher *W**_f_* (1.53 and 0.66 mJ compared with 0.18 mJ) and higher tensile strength (2.93 MPa and 2.23 MPa compared with 1.2 MPa), which is due to the binder action of the polymers incorporated in the particles by co-spray drying. Comparing the PCT-hpc15/PRP and PRP-hpmc10/PCT tablets, the former showed greater toughness (1.53 compared to 0.66 mJ) and tensile strength (2.93 MPa compared to 2.23 MPa). The greater toughness and strength of PCT-hpc15/PRP are partly due to the higher polymer content (15% vs. 10%) and partly due to the higher strength of HPC bonds [36]. The slopes of the curves of the green tablets in the initial linear elastic region (up to 0.4 mm displacement) appear similar for the three drug mixture tablets with physical mixture having lower elastic modulus (258.6 MPa) followed by PCThpc15/PRP (268.3 MPa) and PRP-hpmc10/PCT (295 MPa). In conclusion, the presence of polymer in the CSD/neat tablets, in addition to imposing a barrier to PCT–PRP intimate contact and interaction, results in tablets of acceptable strength (>2 MPa) able to withstand stresses during further processing.

From Figure 9, it appears that the fracture profiles of green (Day 0) and aged (Day 30) tablets show distinct differences which follow two trends. The curves of the aged PCT/PRP and PCT-hpc15/PRP tablets are flatter and located below the respective curves of the green tablets, indicating loss of rigidity and strength, whereas the curve of aged PRP-hpmc10/PCT tablet is an upwards continuation of the green curve to higher breaking loads. The decrease in toughness (0.18 to 0.15 mJ), rigidity (258.6 to 105.6 MPa) and tensile strength (1.20 to 0.86 MPa) of the PCT/PRP tablets are attributed to the reduction in *p*_F_, as explained above. However, the softening of the aged PCT-hpc15/PRP tablets demonstrated by severe loss of Young’s modulus (from 268.3 to 20.9 MPa) and tensile strength (from 2.93 to 1.87 MPa) should be attributed to plasticization of HPC by the moisture absorbed at the RH 75% of stability test [17]. Regarding the hardening of PRP-hpmc10/PCT tablets seen as tensile strength increase (from 2.23 to 3.42 MPa) there are two possible reasons. One is continued plastic deformation and increase in the bonding area due to the absorbed moisture during the stability test [37]. Another reason could be that the absorbed moisture on the polymer layers has become an integral part of the particle surface, effectively increasing the particle volume and decreasing the overall interparticle distance [38].

### 3.7. Raman Spectroscopy of Green and Aged Tablets

To further understand the differences in the mechanical properties, Raman spectra were taken before (Day 0) and after (Day 30) the stability test. These are depicted in Figure 10. Comparison of the spectra of green and aged PCT/PRP tablets does not show changes in the position and intensity of the peaks, which would suggest chemical interaction. Similarly, comparison of the spectra of green and aged PRP-hpmc10/PCT tablets shows only small differences with more characteristics at 1000 cm^−1^, where in the spectrum of the aged tablets a well-formed peak appears, which is underdeveloped in the spectrum of the green tablets. However, comparison of the spectra of green and aged PCT-hpc15/PRP tablets shows several noticeable differences. The former show no peaks or show underdeveloped peaks at positions: 650, 850, 1000, 1350 and 1600 cm^−1^, whereas the latter show distinct peaks which also appear in the spectra of the physical mixture tablets (bottom of Figure 10).

The suppression of the peaks in the green PCT-hpc15/PRP tablet spectra is related to the binder action of the polymers and the associated higher solid fraction (Table 6). Their appearance at the end of the stability test is related to the loss of binder action and associated low solid fraction (Table 6). The higher solid fraction of the green tablets means a denser and smoother surface, and has been proposed that a smoother tablet surface results in reduced light scattering, attenuated signal and suppression of the Raman peaks [39,40]. On the other hand, the low solid fraction of the aged tablets due to swelling of the polymer means more surface pores and more light scattering resulting in appearance of the peaks which are also present in the physical mixture tablet.

### 3.8. Drug Content of Green and Aged Tablets

In Table 8, results of the % drug content analysis for green (Day 0) and aged (Day 30) tablets of the optimized binary mixture and of the physical mixture are presented. It can be seen that in all cases the drug content did not change significantly and ranged between 98.87 and 99.99%.

### 3.9. In Vitro Dissolution Profiles

The polymers employed in this work find use as tablet matrix formers for delayed release at concentrations 15–35%, tablet binders at 2–6% *w*/*w* and as conventional coating film-formers at about 2–4% *w*/*w* [19,41]. Additionally, there are differences in their ability to modify drug release. Vueba et al. found that HPC was not suitable for controlled release matrix tablets while HPMC was advantageous [27]. On the other hand, Kleinebudde (1994) reported that the presence of HPC in pellets accelerated the dissolution of propyphenazone, caffeine and acetaminophen [42]. In light of the above results, and since the content of the polymers in the CSD particles was appreciable, examining the effect of polymer on the release of PCT and PRP from the tablets of the two optimal CSD/neat drug binary mixtures was considered to be of interest.

The release kinetics of the two drugs were examined using the Weibull model. Parameters of the Weibull equation after fitting the model in the dissolution data of the optimized formulations (PCT-hpc15/PRP, PRP-hpmc10/PCT) and the physical mixture (PCT/PRP) are given in Table 9. In all cases, high *R*^2^ values (≥0.992) can be seen, indicating good fitting. For the PCT/PRP tablets the b values were 0.92 and 0.97 for PCT and PRP, respectively, which implies a combined diffusion mechanism. The *t*_d_ values were 22.7 for PCT and 40.9 for PRP, implying relatively fast release of both drugs. For the PCT-hpc15/PRP tablets, the b values were 0.35 for PCT, implying fractal diffusion similar to the percolation cluster and 0.76 for PRP, implying a combined diffusion mechanism. The *t*_d_ values were small, i.e., 1.2 for PCT and 14.3 for PRP, implying fast release from the PCT-hpc15/PRP tablets. For the PRP-hpmc10/PCT tablets, the b values were 0.68 for PCT, implying normal diffusion and 0.77 for PRP, implying a combined diffusion mechanism. The *t*_d_ value for PCT was 58.5, but it was very high for PRP (103.8), implying that HPMC provided a sustained release of PRP.

The above results are visualized in Figure 11, showing in vitro release profiles of PCT (10a) and PRP (10b) of the optimal PCT-hpc15/PRP and PRP-hpmc10/PCT tablets. Results of release from physical mixture tablets of the drugs are also included for comparison. Different behaviors of the CSD/neat tablets are observed relative to the physical mixture tablet. PCT-hpc15/PRP tablet demonstrates significantly faster release than the PCT/PRP tablet, whereas PRP-hpmc10/PCT tablet presents slow diffusion-controlled release. The faster release of the PCT-hpc15/PRP tablet can be attributed to wetting and improved hydrophilicity of the tablet facilitating fluid permeation and dissolution. The slow release of the PRP-hpmc10/PCT tablet and the shape of the release curve imply diffusion-controlled release through the HPMC network on the particle surface.

## 4. Conclusions

This work explored the benefits of the application of co-spray drying for the production of a combination tablet of two incompatible analgesic drugs: paracetamol (PCT) and propyphenazone (PRP). This approach follows two modern strategies: formulation of a fixed-dose-combination of incompatible drugs and continuous manufacturing. The two drugs are prone to eutectic melting when they coexist in a combination tablet. This is due to several reasons: they form eutectic at low temperature of 56 °C that is possible during tableting because of the frictional forces developing at high-speed compaction of the low compressibility drugs; the intimacy of drug molecules during compaction under high compressive forces; and the closeness of the therapeutic and eutectic PCT/PRP ratios. The results of this study show that the achieved surface coverage with the widely used as binders and film formers cellulosic ethers HPMC and HPC increased the melting points of the drugs in the heated mixtures, compared to those in the physical mixtures, and increased the separation of their melting endotherms as well. More importantly, co-spray drying greatly improved the compressibility of the drugs and the strength of the combination tablet, providing at the same time easier detachment from the punch surface and ejectability with anticipated reduction in temperature rise during compaction. The combination of co-spray dried PCT/HPC with neat PRP accelerated the release whereas the combination of co-spray dried PRP/HPMC with PCT slowed down the release of both drugs compared to the physical mixtures. The achieved benefits may contribute towards the formulation of a stable PCT–PRP combination tablet by direct compression.

## Figures and Tables

**Figure 1 pharmaceutics-13-01259-f001:**
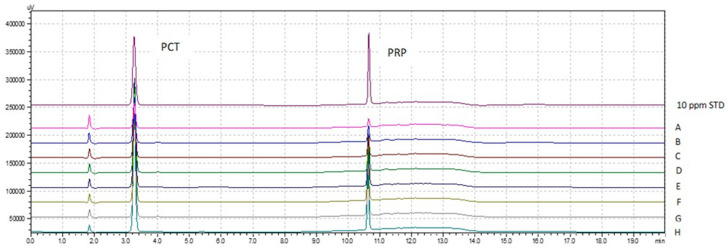
Overlay of representative HPLC-UV chromatographs from analysis of a standard (60:40) mixture of paracetamol (PCT) and propyphenazone (PRP) (10 μg mL^−1^) and chromatographs of dissolution samples drawn at 5, 10, 15, 30, 45, 60, 90 and 120 min (A, B, C, D, E, F, G, H, respectively).

**Figure 2 pharmaceutics-13-01259-f002:**
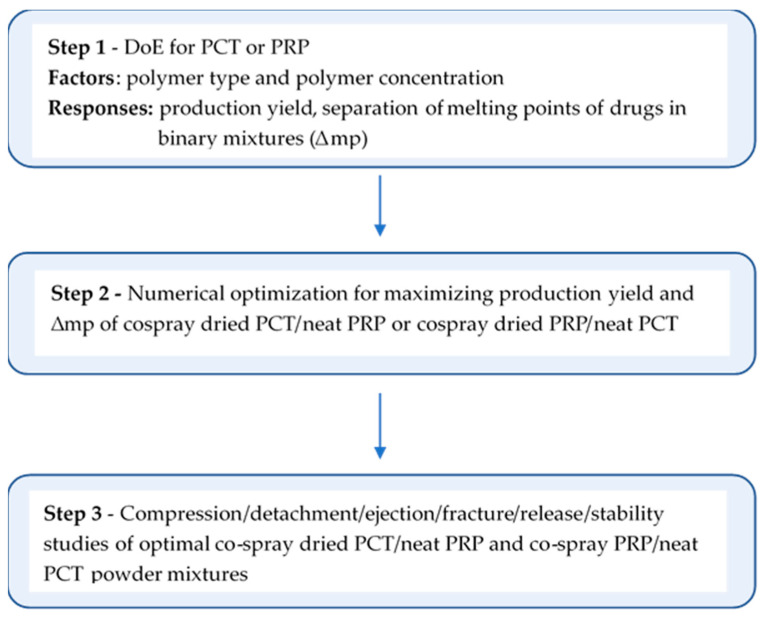
Flow chart of work progress.

**Figure 3 pharmaceutics-13-01259-f003:**
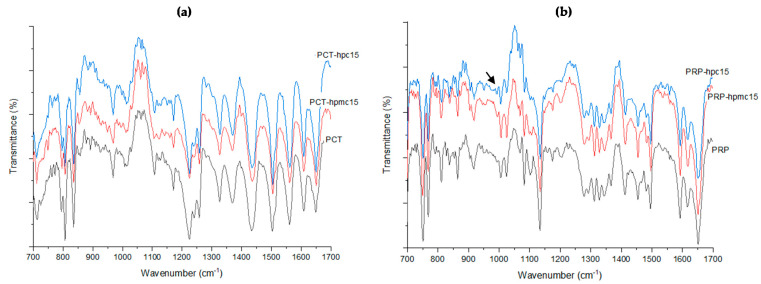
ATR–FTIR spectra of neat PCT (**a**) and neat PRP (**b**) and corresponding co-spray dried batches.

**Figure 4 pharmaceutics-13-01259-f004:**
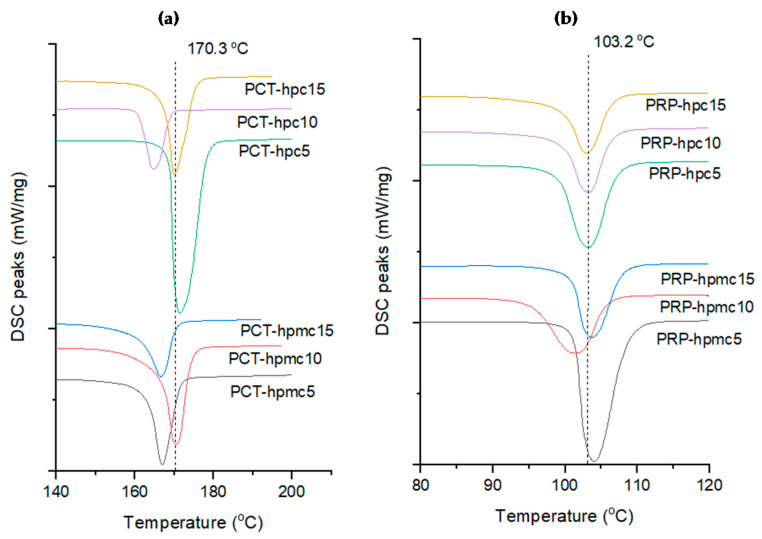
DSC thermograms of (**a**) co-spray dried paracetamol (PCT), and (**b**) co-spray dried propyphenazone (PRP) alone with the two studied polymers (dotted lines correspond to melting points of pure drugs).

**Figure 5 pharmaceutics-13-01259-f005:**
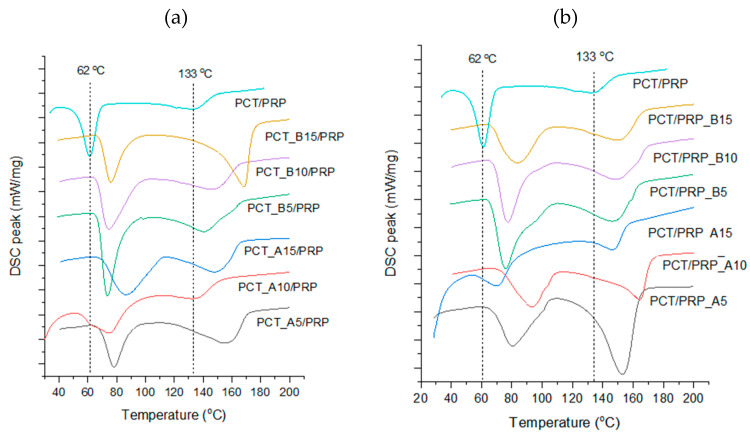
DSC thermograms of binary mixtures of (**a**) co-spray dried PCT/neat PRP, and (**b**) co-spray dried PRP/neat PCT (dotted lines correspond to melting points of the drugs in their physical mixture).

**Figure 6 pharmaceutics-13-01259-f006:**
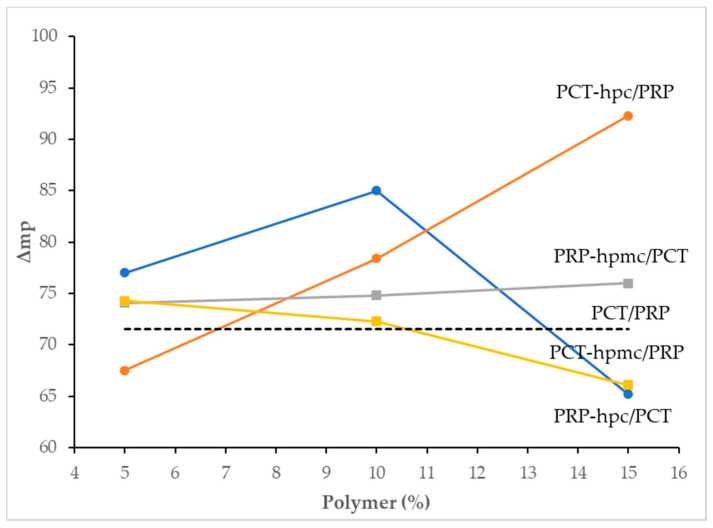
Plots of melting point difference (Δmp) in heated binary co-spray dried/neat mixtures of paracetamol (PCT) and propyphenazone (PRP) with polymer content.

**Figure 7 pharmaceutics-13-01259-f007:**
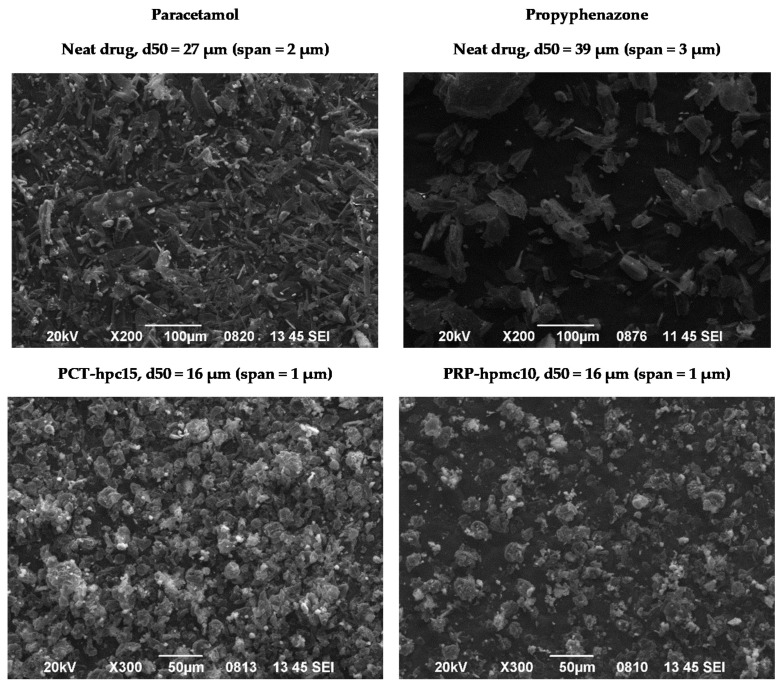
SEM microphotographs of neat drugs and optimal co-spray dried batches of paracetamol with 15% HPC (PCT-hpc15) and propyphenazone with 10% HPMC (PRP-hpmc10).

**Figure 8 pharmaceutics-13-01259-f008:**
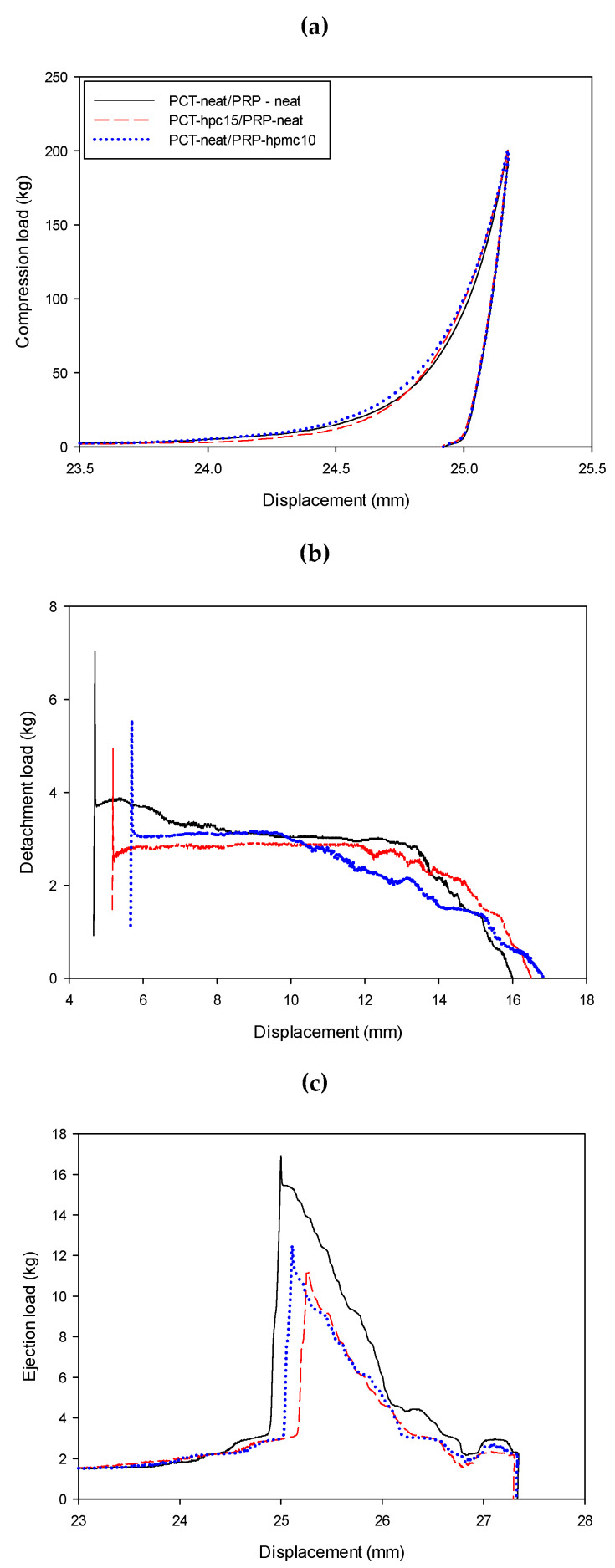
Force–displacement profiles of: (**a**) compression/decompression, (**b**) detachment and (**c**) ejection phases for optimal mixtures of PCT-hpc15 with neat PRP (PCT-hpc15/PRP) and PRP-hpmc10 with neat PCT (PRP-hpmc10/PCT). Profiles of the physical mixture of neat drugs (PCT/PRP) are also shown for comparison.

**Figure 9 pharmaceutics-13-01259-f009:**
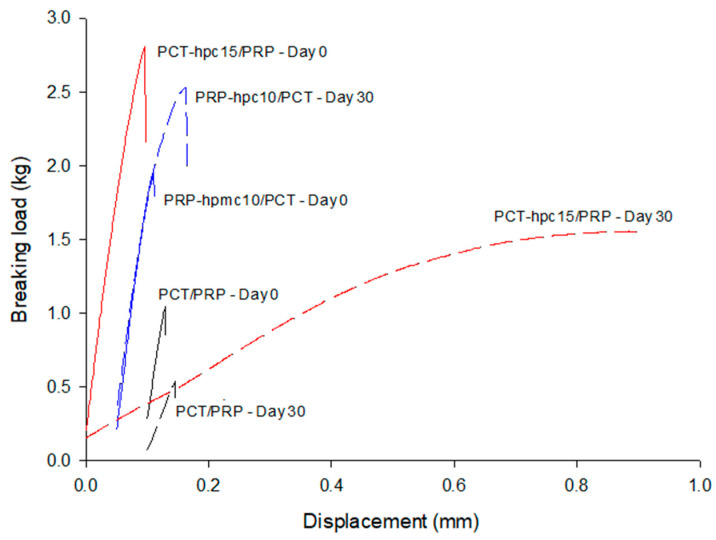
Breaking load against displacement of compression head during diametrical loading for tablets of physical mixture of drugs (PCT/PRP) and CSD/neat drugs at Day 0 and Day 30 (end of stability test).

**Figure 10 pharmaceutics-13-01259-f010:**
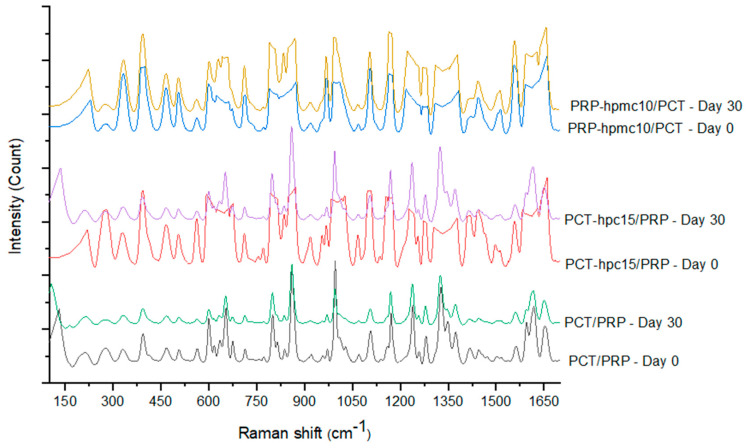
Raman spectra of experimental tablets at the beginning (Day 0) and at the end (Day 30) of the accelerated stability test.

**Figure 11 pharmaceutics-13-01259-f011:**
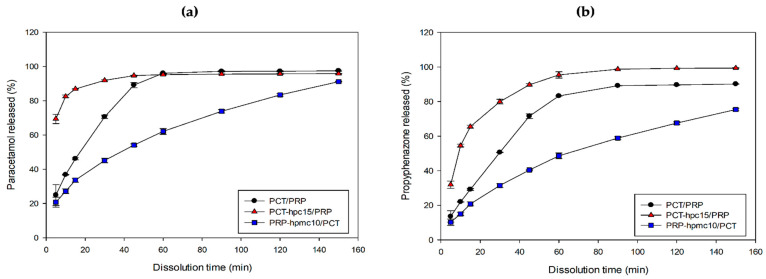
In vitro release profiles of (**a**) PCT and (**b**) PRP from the optimal CSD/neat combination tablets.

**Table 1 pharmaceutics-13-01259-t001:** Factorial experimental design (DoE is described by the first twelve rows after column titles), batch codes, composition, co-spray drying yield and loading (mean ± SD).

Batch Code	Drug/Polymer	Polymer (%)	Yield (%)	Loading (%)
PCT-hpmc5	Paracetamol/HPMC	5	18.6	90.8 ± 1.01
PCT-hpmc10		10	16.1	83.0 ± 0.27
PCT-hpmc15		15	10.7	80.8 ± 1.43
PCT-hpc5	Paracetamol/HPC	5	19.2	99.8 ± 0.14
PCT-hpc10		10	10.5	99.5 ± 0.22
PCT-hpc15		15	9.6	82.4 ± 0.44
PRP-hpmc5	Propyphenazone/HPMC	5	19.8	92.7 ± 0.26
PRP-hpmc10		10	16.8	88.8 ± 0.34
PRP-hpmc15		15	11.0	82.1 ± 0.78
PRP-hpc5	Propyphenazone/HPC	5	18.9	93.8 ± 1.36
PRP-hpc10		10	13.7	86.6 ± 0.84
PRP-hpc15		15	7.8	82.7 ± 1.21
PCT	Paracetamol/neat	n.a.	n.a.	n.a.
PRP	Propyphenazone/neat	n.a.	n.a.	n.a.

n.a., not applicable.

**Table 2 pharmaceutics-13-01259-t002:** Melting points (mp) of co-spray dried drugs alone and in co-spray dried/neat binary mixtures together with mp increase due to the presence of polymers relative to physical mixture of the drugs (both in neat form).

Batch	PCT-CSD	Binary Mixtures of Co-Spray Dried PCT with Neat PRP			
		Co-Spray Dried PCT		Neat PRP	
	mp (°C)	mp (°C)	mp Increase (%)	mp (°C)	mp Increase (%)
PCT-hpmc5	166.9	154.5	17.0	77.7	27.4
PCT-hpmc10	170.5	139.8	5.9	70.8	16.1
PCT-hpmc15	166.4	149.7	13.4	84.5	38.5
PCT-hpc5	171.3	141.3	7.0	73.8	21.0
PCT-hpc10	164.9	152.5	15.5	74.1	21.5
PCT-hpc15	170.1	168.2	27.4	75.9	24.4
**Co-spray-dried sample code**	**PRP-CSD**	**Binary mixtures of PRP–CSD with PCT**			
		**Co-Spray dried PRP**		**Neat PCT**	
	**mp (°C) **	**mp (°C)**	**mp Increase (%)**	**mp (°C)**	**mp Increase (%)**
PRP-hpmc5	104.1	80.2	31.5	154.3	16.9
PRP_hpmc10	101.3	89.3	46.4	163.1	23.6
PRP-hpmc15	103.9	70.5	15.6	146.5	11.0
PRP-hpc5	103.2	75.1	23.1	149.4	13.2
PRP-hpc10	103.2	77.3	26.7	149.6	13.3
PRP-hpc15	103.1	84.5	38.5	150.6	14.1

PCT, paracetamol; PRP, propyphenazone. Binary mixtures had fixed PTC/PRP mass ratio 60:40. Measured mp’s of neat drugs were: for PCT 170.3 °C and for PRP 103.2 °C; in the physical mixtures mp of PCT was 133.0 °C and of PRP 61.0 °C.

**Table 3 pharmaceutics-13-01259-t003:** Statistical equations derived from ANOVA for yield of spray drying of each drug and melting point difference in drugs (Δmp) in mixtures of co-spray dried/neat drugs as a function of polymer type (X1) and polymer content (X2).

Property	Significance of Terms (*p*-Values)			Model Equations in Terms of Actual Components	*R* ^2^	*R*a^2^	*R*p^2^
	X1	X2	X1 × X2				
Production yield for PCT	<0.001	0.013	0.026	HPMC: Y1 = 22.73 − 0.78 × X2 HPC: Y1 = 23.85 − 1.05 × X2	0.971	0.935	0.887
Δmp for co-spray dried PCT/neat PRP	<0.001	<0.001	<0.001	HPMC: Y2 = 81.45 − 1.11 × X2 HPC: Y2 = 54.85 + 2.47 × X2	0.989	0.987	0.982
Production yield for PRP	<0.001	<0.001	0.017	HPMC: Y3 = 24.47 − 0.87 × X2 HPC: Y3 = 24.56 − 1.10 × X2	0.977	0.967	0.945
Δmp for co-spray dried PRP/neat PCT	<0.001	0.032	<0.001	HPMC: Y4 = 73.05 + 0.15 × X2 HPC: Y4 = 78.42 − 0.77 × X2	0.954	0.913	0.874

*R*^2^: coefficient of determination, *R*a^2^: adjusted—*R*^2^, *R*p^2^: predicted—*R*^2^.

**Table 4 pharmaceutics-13-01259-t004:** Numerical solutions for optimization of spray drying production yield and melting point separation (Δmp) obtained from the analysis of the experimental design.

3A. Optimization of Production Yield (%)				
S/N	Batch	Yield (%)	Partial Desirability	Overall Desirability
1	PCT-hpc5	16.37	0.983	n.a.
2	PRP-hpmc5	13.91	0.966	n.a.
3 ^+^	PCT-hpc10	12.60	0.956	0.941
4 ^#^	PRP-hpmc10	10.13	0.934	**0.967**
5 *	PCT-hpc15	8.82	0.919	**0.954**
6	PRP-hpmc15	6.36	0.885	n.a.
**3B.** Optimization of Δmp of co-spray dried PCT/neat PRP mixtures				
S/N	Batch	Δmp (°C)	Partial desirability	Overall desirability
7 *	PCT-hpc15	91.37	0.989	**0.954**
8 ^+^	PCT-hpc10	79.28	0.926	0.941
9	PCT-hpmc10	49.75	0.733	n.a.
**3C.** Optimization of Δmp of co-spray dried PRP/neat PCT mixtures				
S/N	Batch	Δmp (°C)	Partial desirability	Overall desirability
10 ^#^	PRP-hpmc10	74.8	0.999	**0.967**
11	PRP-hpc10	70.9	0.743	n.a.

Numerical solutions having the same “+” or “#” or “*” superscript symbols were used for the calculation for the overall desirability value. n.a., not applicable. Bold characters indicate the highest overall desirabilities corresponding to the numerical solutions for further processing.

**Table 5 pharmaceutics-13-01259-t005:** Results of “in-die” measured parameters for tablets of optimal co-spray dried/neat compositions.

Composition of the Combination Tablet	*W**_c_* (kJ)	*ER* (%)	*F*_det_ (N)	*F*_ej_ (N)
PCT/PRP	569.0 ± 26.9	26.3 ± 3.9	59.8 ± 7.3	162.3 ± 21.4
PCT-hpc15/PRP	664.2 ± 37.9	25.2 ± 3.3	48.9 ± 5.2	117.4 ± 16.2
PRP-hpmc10/PCT	612.6 ± 31.8	29.5 ± 3.9	50.2 ± 5.4	124.1 ± 22.3

*W**_c_*, work of compaction; *ER*, elastic recovery; *F*_det_, force of detachment; *F*_ej_, ejection force.

**Table 6 pharmaceutics-13-01259-t006:** Weight, thickness and solid fraction of tablets before and after the stability test.

Combination Tablet	Weight (mg)		Thickness (mm)		Solid Fraction	
	Day 0	Day 30	Day 0	Day 30	Day 0	Day 30
PCT/PRP	20.35 ± 0.28	20.57 ± 0.34	1.53 ± 0.01	1.80 ± 0.05	0.893 ± 0.032	0.769 ± 0.015
PCT-hpc15/PRP	20.23 ± 0.13	20.78 ± 0.09	1.40 ± 0.01	1.66 ± 0.03	0.955 ± 0.023	0.843 ± 0.012
PRP-hpmc10/PCT	20.29 ± 0.05	20.32 ± 0.05	1.44 ± 0.01	1.44 ± 0.02	0.940 ± 0.008	0.946 ± 0.005

**Table 7 pharmaceutics-13-01259-t007:** Results of fracture test of tablets of optimal combinations: solid fraction, work of failure, Young’s modulus and tensile strength.

Combination Tablet	*W**_f_* (mJ)		*E* (MPa)		*TS* (MPa)	
	Day 0	Day 30	Day 0	Day 30	Day 0	Day 30
PCT/PRP	0.18 ± 0.05	0.15 ± 0.02	258.6 ± 4.8	105.6 ± 1.7	1.20 ± 0.33	0.86 ± 0.16
PCT-hpc15/PRP	1.53 ± 0.09	7.99 ± 2.08	268.3 ± 5.9	20.9 ± 1.5	2.93 ± 0.03	1.87 ± 0.32
PRP-hpmc10/PCT	0.66 ± 0.08	2.04 ± 0.15	295.0 ± 6.4	214.1 ± 4.8	2.23 ± 0.01	3.42 ± 0.17

*W**_f_*, work of failure; *E*, Young’s modulus; *TS*, tensile strength.

**Table 8 pharmaceutics-13-01259-t008:** Results of drug content analysis of tablets before and after the stability test.

Combination Tablet	% PCT Content		% PRP Content	
	Day 0	Day 30	Day 0	Day 30
PCT/PRP	99.99 ± 0.01	99.92 ± 0.03	99.99 ± 0.01	98.87 ± 0.08
PCT-hpc15/PRP	99.99 ± 0.01	99.95 ± 0.02	99.98 ± 0.01	99.12 ± 0.11
PRP-hpmc10/PCT	99.99 ± 0.01	99.91 ± 0.02	99.95 ± 0.02	99.87 ± 0.07

**Table 9 pharmaceutics-13-01259-t009:** Weibull equation parameters (mean ±SD, *n* = 3) for the dissolution of paracetamol and propyphenazone form the optimized formulations and the physical mixture.

Combination Tablet	Parameters for PCT			Parameters for PRP		
	*b*	*t* _d_	*R* ^2^	*b*	*t* _d_	*R* ^2^
PCT/PRP	0.92 ± 0.01	22.7 ± 0.6	0.959	0.97 ± 0.01	40.9 ± 0.5	0.978
PCT-hpc15/PRP	0.35 ± 0.01	1.2 ± 0.1	0.932	0.76 ± 0.01	14.3 ± 0.6	0.992
PRP-hpmc10/PCT	0.68 ± 0.01	58.5 ± 2.9	0.991	0.77 ± 0.02	103.8 ± 2.6	0.999

## Data Availability

Not applicable.

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
