# Peer review of "Co-Spray Drying of Paracetamol and Propyphenazone with Polymeric Binders for Enabling Compaction and Stability Improvement in a Combination Tablet"

_pharmaceutics, 2021, doi:10.3390/pharmaceutics13081259_

Round 1

Reviewer 1 Report

The manuscript entitled “Co-spray drying of paracetamol and propyphenazone with polymeric binders for stability and compaction improvement in a combination tablet” deals with a complete characterization study of drug/polymer suspensions for achieving the optimal tablets composed by two drugs that were also characterized and stability studies were carried out. So, PCT and PRP were co-spray dried separately with HPMC and HPC to create a polymer barrier around the drug particles to minimize contact and interaction between the two drugs and to produce spherical particles with uniform particle size and improved compressibility, thus enabling formation of a direct compression combination tablet.

I found the manuscript very complete in terms of characterization studies carried out and discussion of the results obtained. It is very well written and correctly justified.

However, there are still topics and small mistakes that should be improved by authors:

Major fixes:

By combining the two drugs with the HPC polymers and mainly with HPMC, these polymers could modify the release rate of the drugs under in vivo conditions. Have you carried out drug release studies, even if only in vitro, in which you have compared the release data of drugs obtained from tablets with polymers versus tablets without polymers as in article 25 of your references?

To study the possible interactions between drugs and polymers, did you consider the X-ray diffraction technique as in article 25 of your bibliography? Why didn't you carry out that technique whose results are usually much easier to read than ATR-FTIR Spectroscopy?

I think that the conclusions should be elaborated again because they are incomplete because you have not specifically collected the best drug / polymer combination obtained to make the most optimal tablets, you have not even talked about polymers in the conclusions.

Minor fixes:

I have detected, like in more sections, several orthographic mistakes:

In introduction, line 50, authors should delete “for example”.

In section 2.1, line 123, change U.K. for UK as line 119; line 125, I think it should be “HPLC analysis”.

In several section (2.2.1; 2.2.2) you write ml. Correct to mL.

Figure 1 looks a bit blurry because it is based on enlarged text. Can you change it to make it look better?

In Table 1 and 3 put n.a. and its meaning (not available or not applicable) in the lower text that helps to know the meaning of the abbreviations.

In section 3.4 line 438 and 439. I think the authors meant PRP-hpmc15 and PTC-hpc10, right?

In section 3.6.1 line 481 change “Ta-ble 4” to Table 4.

In References:

I think that the number of citations, excluding those of the authors of the manuscript themselves, are a little low.

Authors should write all doi in the same way, i.e. o doi: 10. ... or doi: https: //doi.org/10. ...

5. Al-Zoubi, N.; Gharaibeh, S.; Aljaberi, A.; Nikolakakis, I. Spray Drying for Direct Compression of Pharmaceuticals. Processes 2021, 9(2), 267, doi:10.3390/pr9020267.

Cite 7 is a book, not an article. Write correctly.

10. Vanhoorne, V.; Van Bockstal, P.-J.; Van Snick, B.; Peeters, E.; Monteyne, T.; Gomes, P.; De Beer, T.; Remon, J.P.; Vervaet, C. Continuous manufacturing of delta mannitol by cospray drying with PVP. International Journal of Pharmaceutics 2016, 501, 139-147, doi: 10.1016/j.ijpharm.2016.02.001.

Cite 19. Have not authors been able to consult a more current edition of the USP Pharmacopoeia?

27. Partheniadis, I.; Toskas, M.; Stavras, F.-M.; Menexes, G.; Nikolakakis, I. Impact of Hot-Melt-Extrusion on Solid-State Properties of Pharmaceutical Polymers and Classification Using Hierarchical Cluster Analysis. Processes 2020, 8(10), 1208, doi:10.3390/pr8101208.

30. Köster, C.; Pohl, S.; Kleinebudde, P. Evaluation of Binders in Twin-Screw Wet Granulation. Pharmaceutics 2021, 13(2), 241, doi:10.3390/pharmaceutics13020241.

31. Nokhodchi, A.; Javadzadeh, Y. The effect of storage conditions on the physical stability of tablets. Pharmaceutical Technology Europe 2007, 19, 20-26.

34. Wang, H.; Mann, C.K.; Vickers, T.J. Effect of Powder Properties on the Intensity of Raman Scattering by Crystalline Solids. 754 Appl. Spectrosc. 2002, 56, 1538-1544, doi:10.1366/000370202321115779.

Author Response

Please find our respons in the attached file.

Reviewer 2 Report

The manuscript (pharmaceutics-1317781) entitled "Co-spray drying of paracetamol and propyphenazone with polymeric binders for stability and compaction improvement in a combination tablet" provides sound discussion with a reasonable set of experiments. Indeed, the manuscript required major revision for further improvement in its quality. Authors are advised to consider the following suggestions.

  1. It is suggested to incorporate detailed investigation on in vitro drug release profile of the developed combination products in simulated gastric fluid,  simulated intestinal fluid, and official dissolution media recommended for the individual drugs.
  2. Authors should investigate the influence of different formulation compositions on its in vitro drug release profile. 
  3. Compare the similarity factors of developed formulations compared to marketed products. 
  4.  It is advised to observe the release kinetics of developed combination products compared to the conventional formulation.
  5. Authors are advised to incorporate the stability data of at least 3 months. The 1-month period is very short to find any interactions, degradation, or other stability issues.
  6. It is also advised to determine the %drug content of the developed combination tablets at different time intervals during stability study. 
  7. Provide the detail of stability indicating method to determine %drug content or drug degradation during stability study as supplementary information.  

Author Response

Please find our response in the attached file

Reviewer 3 Report

In this article, authors have tried to improve the stability and compaction properties of paracetamol (PCT) and propyphenazone (PRP) using co-spray drying technology in combined tablet dosage form. The prepared formulations were characterized physicochemically and evaluated for stability. The manuscript is written nicely but its practical applications are limited. Several combinations of PCT and PRP are already available in solid dosage form in which the stability of both drugs is not an issue. We cannot consider the superiority of present formulations over marketed formulations based on stability improvement. The biological studies are important to show the superiority of present formulations. The publication of this manuscript is premature at this stage. Authors are advised to include pharmacodynamics and pharmacokinetic data of present formulations in order to prove their superiority over marketed formulations. Due to lack of biological data, I must recommend its rejection for the publication.

Author Response

Please find our response in the attached file.

Reviewer 4 Report

The paper entitled “Co-spray drying of paracetamol and propyphenazone with polymeric binders for stability and compaction improvement in a combination tablet” offers a  proof-of concept about the use of design of experiment (DoE) approach to optimize the polymer concentration on the yield of spray drying and melting point separation of heated binary mixtures.

Overall, the results obtained by an extensive number of experiments well support the conclusions and the authors nicely present them. Either the abstract or the introduction well describe the system and its application.

However, it would be particularly appreciated to have a section where the DoE is described in a relatively easy way, so that even a “non-expert” reader can understand the importance of this tool, which is particularly exploited in Pharma Company, while it is rare to find it in academia.
For this region, I suggest adding a general introduction about the DoE directly in the introduction.
Specifically, a section that clearly elucidates the design employed with its description and the reason why the authors select this specific design is missing. Inexpert reader could not understand the importance of such statistical approach.
I recommend writing a small paragraph where all these aspects are elucidate.
Moreover, the experimental and the results section have to be implemented to improve the quality of the work.
• Specifically, the DoE design has to be described. The Statistical analyses to evaluate the outcome is not described and any information is given about how the response were obtained and how the DoE was made. I would add at least the model summary (S; R-sq; R-sq adj and R-sq pred) for each DoE.
• Did the author use a mixture design? Was an extreme vertex? How many degree of freedom? Was the DoEs augmented with central or axial points? How the authors did analyzed the DoE? Did they use a special cubic, quadratic or special quadratic equation to fit the model? Please, describe all the parameters and the statistics of your DoE approach. A Design summary is missing. Please add the following information: Runs, Blocks, Base design, Replicates and Central points.
Furthermore, accordingly with the fig 1, seams that the authors analyzed the same DoE using two different outcome: (1) on the yield of spray drying and (2) on the separation of melting endotherms (Δmp) of the drugs.
Why the D was not optimized for the two outcomes together? The DoE can analyzed by selecting multiple responses. An Overlaid Contour Plot could be used to visually identify an area where the predicted means of more response variables are in an acceptable range. In your case, optimal variable values for one response may be far from optimal for another response. Overlaid contour plots can allow you to visually identify an area of compromise among the various responses.
I also suggest adding either a response trace plot or a contour plot that clearly show graphically how the variables affect the outcomes.

From my side the quality of the work does not meet the quality criteria of the journal and it can be published only after this revision.

Author Response

(The authors gave the same response as above.)

Round 2

Reviewer 1 Report

After all changes in the manuscript, I think

that it is suitable to be published.

Reviewer 2 Report

The revised manuscript (pharmaceutics-1317781) entitled "Co-spray drying of paracetamol and propyphenazone with polymeric binders for stability and compaction improvement in a combination tablet" improved well after incorporating the given suggestions. In my opinion, the present form of the manuscript should be considered for publication in Pharmaceutics. 

Reviewer 3 Report

The authors have addressed previous concerns. The manuscript can be accepted in its current form.